# Design of Novel Perovskite-Based Polymeric Poly(l-Lactide-Co-Glycolide) Nanofibers with Anti-Microbial Properties for Tissue Engineering

**DOI:** 10.3390/nano10061127

**Published:** 2020-06-07

**Authors:** Aleksander Góra, Lingling Tian, Seeram Ramakrishna, Shayanti Mukherjee

**Affiliations:** 1Center for Nanofibers and Nanotechnology, Faculty of Engineering, Engineering Drive, National University of Singapore, Singapore 117576, Singapore; aleksander.gora@sfm.sg (A.G.); lingling@nus.edu.sg (L.T.); seeram@nus.edu.sg (S.R.); 2Sport & Fashion Management Pte. Ltd., 6 Shenton Way, OUE Downtown 18-11, Singapore 068809, Singapore; 3The Ritchie Centre, Hudson Institute of Medical Research, Clayton, Victoria 3168, Australia; 4Department of Obstetrics and Gynecology, Monash University, Clayton, Victoria 3168, Australia

**Keywords:** electrospinning, nanofiber, tissue engineering, perovskite, anti-microbial, nanomaterial

## Abstract

There is a growing need for anti-microbial materials in several biomedical application areas, such are hernia, skin grafts as well as gynecological products, owing to the complications caused by infection due to surgical biomaterials. The anti-microbial effects of silver in the form of nanoparticles, although effective, can be toxic to surrounding cells. In this study, we report, for the first time, a novel biomedical application of Ag_0.3_Na_1.7_La_2_Ti_3_O_10_-layered perovskite particles, blended with poly(L-lactide-co-glycolide) (PLGA), aimed at designing anti-microbial and tissue engineering scaffolds. The perovskite was incorporated in three concentrations of 1, 5, 10 and 15 w/w% and electrospun using dimethylformamide (DMF) and chloroform. The morphology of the resultant nanofibers revealed fiber diameters in the range of 408 to 610 nm by scanning electron microscopy. Mechanical properties of perovskite-based nanofibers also matched similar mechanical properties to human skin. We observed impressive anti-microbial activity, against Gram-negative, Gram-positive bacteria and even fungi, to Ag_0.3_Na_1.7_La_2_Ti_3_O_10_ in powder as well as nanofiber-incorporated forms. Furthermore, cytotoxicity assay and immunocytochemistry revealed that perovskite-based nanofibers promoted the proliferation of human dermal fibroblasts whist maintaining normal cellular protein expression. Our study shows that perovskite-nanofibers have potential as scaffolds for biomedical applications with anti-microbial needs.

## 1. Introduction

Anti-microbial materials are gaining more and more interest in the scientific community [1]. Development of antibiotic resistant strains as well as new insight into wound healing mechanisms on a cellular level have allowed for the development of new materials that not only protect wounds from infection, but actively fight pathogens around the wound and stimulate tissue for self-regeneration. Those materials include various polymeric structures with proteins, metal oxides, metal nanoparticles particles as well as antibiotics and growth factors. The rising number of antibiotic resistant microbes has forced researchers to find materials that can limit the use of antibiotics in wound management [2,3,4].

Perovskites are a group of ceramic materials based on a crystallographic structure of calcium titanium oxide (CaTiO_3_). The standard cell unit can be altered during synthesis with the creation of double or layered structures [5]. In recent years, a number of materials with perovskite structures have been proposed as novel materials due to its unique electric and magnetic properties. So far, the biggest interest of use of these new structures lie within solar cell technology, due to much higher efficiency than traditional materials [6,7]. Synthesis of perovskites allows for construction of structures with various properties and arrangements of atoms, with possible substitution of atoms in the ceramic lattice and production in the sub-micron range of particles. Recently, anti-microbial behaviors of some custom synthesized perovskites have been proposed [8]. However, the application of perovskite in tissue engineering has not been explored.

Titanium oxide (TiO_2_) is very well-known material with anti-microbial properties. Commercially it is being used in skincare as an excellent UV absorber, as well as a pigment and thickener of the product, and in industry as a catalyzer [9]. Titanium oxide as an anti-microbial nanoparticle has been proven to work against most Gram-positive and Gram-negative bacteria [10,11]. Furthermore, silver is a very well-known anti-microbial material with a long history of usage dating back to the twentieth century. Because it was not extensively used in the twentieth century, bacteria and fungi did not gain resistance against silver [12].

The complete mechanism of the reaction between silver and bacteria cells is not completely known so far; nevertheless, it has been suggested that silver ions and particles are accumulated on multiple sites on the cell membrane, changing its permeability, and they may also interact with DNA and cause enzymatic imbalance. This multisite reaction is the significant advantage of silver nanoparticles over traditionally used anti-microbial agents [13]. Studies on AgNP’s cellular intake showed that in higher amounts, they are toxic due to the accumulation of nanoparticles within the cell structures [14,15]. Studies have revealed that AgNPs can induce toxicity to human dermal fibroblast, often by disruption of the cell membrane, cytoskeleton and energy metabolism pathways that ultimately lead to cell cycle arrest [16,17,18,19]. However, given its anti-microbial potential, efforts to reduce the toxicity of Ag-based biomaterials on human cells has continued [20], and several impressive material science strategies, such as reduction of nanoparticles size [21], have been applied. Application of biomaterial advances to control microbial growth through the release of anti-microbial agents such as Ag continues to be a prime area of research across many fields [22,23,24]. This study explores the use of anti-microbial perovskite materials containing Ag and hypothesizes that such materials might have the potential to limit cell toxicity but preserve the anti-microbial properties of wound dressing. During the synthesis process, atoms in the perovskite lattice can be substituted with silver atoms and released in the form of ions into the wound to ensure anti-microbial activity. It is already known that perovskites are efficient against Gram-negative and Gram-positive bacteria with minimal cytotoxicity [25,26,27,28,29]. However, their efficacy in the form of tissue engineering scaffolds remains unknown.

The electrospinning technique is one of the methods of production of nanofibrous non-woven polymeric or ceramic materials with high scaling up potential. This technology is capable of production of fibers with diameters in the range of 50–1000 nm, and therefore the product of electrospinning is typically named as nanofibers [30]. They have been widely used in several tissue engineering applications owing to their biocompatibility to act as a scaffold for cellular activities. The electrohydrodynamic process of electrospinning utilizes a polymer solution in organic or non-organic solvent as well as polymeric blends or polymer mixtures with additional proteins, particles, antibiotics or precursors [31]. Nanofibers are especially interesting materials in wound management because of their high porosity, high strength, controllable composition, ability to release drugs or other compounds, very high surface to volume ratio and permeability to vapor. Nanofibrous wound dressings are able to ensure sufficient protection against microbes due to possible incorporation of active agents and maintaining suitable microclimate between wound and dressing [32].

Electrospinning has also emerged in the development of three-dimensional (3D) in vitro models to understand the behavior of skin cells such as keratinocytes [33]. Recent progress on skin grafts using nanofibers has shown improved cell response, while additional 3D structures were introduced before nanofiber spinning by laser assisted 3D printing [34]. Moreover, creating dressings from biodegradable polymeric matrixes gives additional control over the release of anti-microbial agents and allows dressing to be used internally to prevent deep surgical site infections. Poly(lactide-co-glycolide), also called PLGA, is a biodegradable polymer that undergoes hydrolytic degradation in living organism with harmless degradation products. By controlling the ratio of lactide to glycolide monomers, degradation time of this polymer can be controlled within a time period of 4 weeks up to 2 years [35,36].

This work summarizes the synthesis of layered perovskite Na_2_La_2_Ti_3_O_10_ by solid state synthesis and describes substitution of sodium atoms with silver in order to create Ag_0.3_Na_1.7_La_2_Ti_3_O_10_ perovskite. Herein, we report, for the first time, the fabrication of a nanofibrous tissue engineering scaffold enriched with perovskites. The prepared particles were incorporated in PLGA nanofibrous scaffolds by the method of electrospinning. Previous studies have reported the preparation [28,29] and incorporation of perovskites in nanofibers [37]. Perovskites have been shown to incorporate very well within nanofibers fabricated by electrospinning [37,38,39,40,41]. However, the potential of perovskite-based nanofibers in tissue engineering application as anti-microbial scaffolds for wound healing has never been explored. Herein, modified perovskite nanofibrous scaffolds were characterized by scanning electron microscopy (SEM), Fourier-transform infrared spectroscopy (FTIR) and universal tensile machine (UTS). Anti-microbial properties were evaluated, against the Gram-positive and Gram-negative bacteria strains *Staphylococcus sarophyticus*, *Klebisiella pneumoniae*, *Pseudomonas aeruginosa* and *Escherichia coli* to investigate the potential of utilizing these nanofibrous scaffolds as novel wound dressings. The scaffolds were assessed for their biocompatibility by human dermal fibroblasts (HDF) by the AlamarBlue^®^ assay. Our study shows that such nanostructured scaffolds have high anti-microbial properties, are non-toxic to human cells and also support their normal cellular protein expression. To the best of our knowledge, this kind of anti-microbial nanocomposite with perovskites has not been reported before. This study presents a novel biomedical application of perovskite-based nanomaterials in generating scaffolds for tissue engineering and provides the first evidence of their anti-microbial and cytocompatibility profiles in the form of electrospun nanofibers.

## 2. Materials and Methods

### 2.1. Materials

Poly(L-lactide-co-glycolide) with an L:G ratio of 75:25 (B6007-1, PLGA 75/25), with an inherent viscosity of 0.55–0.75, was purchased from LACTEL Absorbable Polymers (Durect, Birmingham, AL, USA). Dimethylformamide (DMF), chloroform (CF), phosphate buffer saline (PBS), sodium carbonate (E5460), lanthanum (III) nitrate (238554), titanium oxide (634662), silver nitrate (209139), Dulbecco’s Modified Eagle’s Medium (DMEM) and penicillin–streptomycin antibiotic solution were purchased form Sigma-Aldrich Co. (St. Louis, MO, USA) and used without further purification. Fetal bovine serum (FBS) and trypsin (10×) were purchased from Gibco (Thermo-Fisher Scientific, Waltham, MA, USA), and AlamarBlue^®^ BUF012B cell viability assay reagent was purchased from AbD Serotec (Kidlington, UK). Human dermal fibroblasts (HDFs) were purchased from ATCC (PCS-201-012, Manassas, VA, USA) and used for the biocompatibility studies. Bacterial and fungi strains used for the experiments were purchased from ATCC (Manassas, VA, USA). Four Gram-positive strains, namely *Staphylococcus saprophyticus* (strain codes: 15305, BAA-750, 49907, 49453) and four Gram-negative strains, namely *Klebsiella pneumoniae* (DM4299), *Escherichia coli* (16027R) and *Pseudomonas aeruginosa* (23155 and 4299) were used for the antibacterial evaluation.

### 2.2. Synthesis of Perovskite Particles

Layered perovskite with Ruddlesden–Popper (R–P) phase material Na_2_La_2_TiO_3_O_10_ was synthesized by the solid state synthesis method by mixing ingredients in molar weight and subjecting the mixture to high temperature [42,43]. As described in Scheme 1A, lanthanum nitrate (La(NO_3_)_3_), titanium oxide (TiO_2_) and sodium carbonate (Na_2_CO_3_) were mixed in a molar ratio of 2:3:1.6. An excess of sodium carbonate of 60% (mol) was add intentionally in order to compensate mass loss during synthesis. Mixed powders were placed in alumina crucibles and dried at 105 °C for 10 h in air. Next, the mixture was calcined at 500 °C for 10 h and heated to 1000 °C for 5 h. After synthesis, the material was washed in distilled water and dried at 105 °C for 10 h. To achieve ion exchange, 1 mol of prepared Na_2_La_2_TiO_3_O_10_ powder was mixed with 0.3 mol of AgNO_3_ in the presence of distilled water (10% suspension) and heated to 60 °C on a magnetic stirrer for 6 h in a dark vessel. After, the ion exchange product was filtered, washed with distilled water and dried at 105 °C for 10 h.

### 2.3. Electrospinning of Perovskite-Incorporated Fibers

Nanofibers in this work were prepared by the electrospinning method from a mixture of PLGA 75/25 polymer and perovskite compound, as described in Scheme 1B. Then 1%, 5%, 10% and 15% perovskite were added to the mixture of chloroform and DMF with a ratio of 1:3 and stirred for 24 h by magnetic stirrer, followed by ultrasound stirring (FB505, Fisher Scientific, Hampton, NH, USA) for 20 min in 30 s intervals. Next, 25% (w/v) of PLGA75/25 polymer was added and stirred for 12 h with a magnetic stirrer in order to completely dissolve the polymer. The solution was mixed again with an ultrasound stirrer prior to electrospinning. Electrospinning equipment for this study consisted of a high-voltage power supply (Gamma High Voltage Research, Ormond Beach, FL, USA), syringe pump (KDS100, KD Scientific, Holliston, MA, USA) and a flat collector. The prepared polymer solution was placed in a 5 mL syringe, and a blunt needle with diameter of 0.51 mm (25 G, Dickinson and Company, NJ, USA) was used as electrode. The whole solution was electro spun under 13.7–14.0 kV voltage applied at a distance of 13–14 cm from the collector with a polymer flow of 1.2 mL/h. The prepared scaffolds were finally placed in a vacuum chamber for 24 h prior to characterization in order to remove solvent residues. Sterilization of scaffolds before cell culture and anti-microbial tests was done by placing materials under a UV lamp for 30 min and keeping them in a sealed sterile environment.

### 2.4. Mechanical Characterization of Electrospun Nanofibers

A scanning electron microscope (FESEM Hitachi S-4300, Ibaraki, Japan) was used to examine and measure diameters of prepared scaffolds. Scaffolds were sputter coated with gold (JOEL JFC-1 200, Tokyo, Japan) before analysis. Diameters of the fibers were measured and analyzed by ImageJ Software (ver. 1.50e, National Institutes of Health, Bethesda, MD, USA). One hundred measurement per sample were performed, and average value as well as standard deviation were taken under further consideration. Mechanical properties of scaffolds were calculated from results obtained from a universal tensile machine (Instron 5943, Norwood, MA, USA). Each sample was cut to a rectangular shape (10 × 20 mm) and placed in a 250 N load cell. The thickness of the scaffold sheet was measured with a micrometer (Multitoyo, Kanagawa, Japan) in three random places and average value was taken under consideration. The thicknesses of samples varied between 80 and 100 μm. UTS measurements were done with a rate of 10 mm/min until breakage. Young’s modulus (YM) for subsequent samples was calculated from stress–strain curves by applying linear fit to curve in the region of 1–5% of elongation. Ultimate tensile stress and ultimate tensile strain were taken as maximum values from the stress–strain curve.

Hydrophilic properties of the materials were measured by water contact angle (VCA, Optima Surface Analysis, AST Products, Billerica, MA, USA). Average values from 3 measurements per sample were taken and presented together with SEM analysis. For each measurement, 10 μL of ultra-pure water was placed on the surface of the scaffold, and a picture was taken within 10 s. Fourier transform infrared spectroscopy (FTIR) was conducted using an Avatar 380 FTIR spectrometer (Thermo Nicolet, Waltham, MA, USA). A range of 400–4000 cm^−1^ was scanned with a resolution of 2 cm^−1^. X-ray diffraction (Panalytical MRD X-ray Diffractometer, Spectris, UK) and particle size measurements (Zetasizer Nano S; Malvern Instruments Ltd., Malvern, UK) were conducted. XRD measurements were done using Cu-Kα radiation (40.0 kV, 30.0 mA) in a continuous scan between 5.0 and 65 degrees with a speed of 2 degree per minute. Zeta-sizer samples were prepared by placing 100 mg of powder in 1 mL of distilled water, and particle distribution was measured at 25 °C with a set dispersant RI of 1.330.

### 2.5. Anti-Microbial Properties of the Nanofibers

Anti-microbial activities of nanofibrous scaffolds were performed on four Gram-negative (*K. pneumonia* (DM4299), *E. coli* (16027R), *P. aeruginosa* (23155 and 4299)) and four Gram-positive (*S. saprophyticus* (15305, BAA750, 49907, 49453)) strains as well as two species of fungi (*C. albicans* (CA1976, CA2672)). All cultures were grown overnight in tryptic soy agar at 37 °C. Mueller–Hinton agar (MHA) containing 30.0% beef infusion, 1.75% casein hydrolysate, 0.15% starch and 1.7% agar with neutral pH at 25 °C was used. Colonies were inoculated in turbidity of 0.5 McFarland standard and swabbed uniformly across the agar surface. Scaffolds of 0.5 × 0.5 cm in size were placed on the surface of agar plates and incubated for 24 h. Non-substituted and substituted perovskite material was dissolved in water in concentration of 2 mg of powder in 1 mL of distilled water. Then, 100 μL of this solution was placed on the MHA plate surface. All experiments were performed in duplicate, and measurements of bacteria and fungi free zones were done after 24 h of incubation at 37 °C.

### 2.6. Culture of Human Dermal Fibroblasts

HDF were cultured in media composed of DMEM with 10% FBS and 1% antibiotic mixture. Cells were grown in 75 cm^2^ flasks for 6 days in an incubator under humidified atmosphere at 37 °C and 5% CO_2_ until 70% confluence. Culture medium was replaced every 3 days. Subsequently, cells were trypsinized, counted and used for further experiments. HDF were cultured on the surfaces of the scaffolds prepared especially for this experiment. Scaffolds were electrospun on glass cover slides and sterilized under UV light for 2 h. Glass slides (15 mm in diameter) were placed in 24 well plates with clean glass slides used as the control group (TCP). In order to secure polymeric scaffolds on the surface of the glass slip, metal rings were placed on top of every sample. Scaffolds in well plates were washed thrice with phosphate-buffered saline (1 × PBS) and immersed in complete culture medium. Then, prepared material was placed in an incubator at 37 °C for 24 h. About 10,000 cells were placed in the each well for viability testing and cultivated for 8 days with the AlamarBlue^®^ assay performed every second day. A calibration curve was calculated by seeding 10,000, 20,000, 40,000, 80,000, 160,000 and 320,000 cells/well and performing the AlamarBlue^®^ assay after 12 h. Number of cells was calculated by substituting results to a linear fitting equation from the calibration curve. The AlamarBlue^®^ assay is commonly utilized to quantitatively measure viability of human and animal cell lines. Those assays were performed at four time points of the experiment (2, 4, 6 and 8 days). During test day, 10% of AlamarBlue^®^ in complete media was used to replace medium in the well and incubated for 4 h at 37 °C. Ten samples of volume of 100 μL from each incubated well were transferred to a 96 well plate. Readings from each well were performed by spectrophotometric plate reader (Varioscan Flash, Thermo Fisher Scientific, Waltham, MA, USA) with excitation and emission length of 545 nm and 590 nm, respectively. Test viability in the AlamarBlue^®^ assay is based on reduction of resazurin to resorufin, which shifts excitation wavelength towards red. In order to minimize errors in measurement, two wells were filled with blank assay and incubated together with scaffolds, and measured values were deducted from sample measurements.

### 2.7. Anti-Microbial Properties of the Scaffolds

Anti-microbial activity of nanofibrous scaffolds were performed on four Gram-negative (*K. pneumonia* (DM4299), *E. coli* (16027R), *P. aeruginosa* (23155 and 4299)) and four Gram-positive (*S. saprophyticus* (15305, BAA750, 49907, 49453)) strains as well as two species of fungi (*C. albicans* (CA1976, CA2672)). All cultures were grown overnight in tryptic soy agar at 37 °C. Mueller–Hinton agar (MHA) contained 30.0% beef infusion, 1.75% casein hydrolysate, 0.15% starch and 1.7% agar with natural pH at 25 °C. Colonies were inoculated in turbidity of 0.5 McFarland standard and swabbed uniformly across the agar surface. Scaffolds of 0.5 × 0.5 cm in size were placed on the surface of the agar plate and incubated for 24 h. Non-substituted and substituted perovskite material was dissolved in water in concentrations of 2 mg of powder in 1 mL of distilled water. Then, 100 μL of this solution was placed on the MHA plate surface. All experiments were performed in duplicate, and measurements of bacteria and fungi free zones were done after 24 h of incubation at 37 °C.

### 2.8. Confocal Microscopy of Human Dermal Fibroblast on Nanofibers

Confocal laser scanning microscopy (LSCM) was performed on scaffolds on day 8 of cell cultivation with the LSM-700 microscope system (LSM-700, Carl Zeiss, Oberkochen, Germany). Firstly, scaffolds were washed with complete media twice at 37 °C. Then, 4% of formaldehyde (Sigma Aldrich Co., Saint Louis, MI, USA) was added in an amount of 200 μL per well and kept for 20 min at −20 °C. Scaffolds were subsequently raised twice with PBS at room temperature. Next, 0.1% triton X 100 (Sigma-Aldrich Co., Saint Louis, MI, USA) solution was added to wells and kept for 90 s and washed again with PBS trice. Then, 3% bovine serum albumin (BSA) was next added to each well and kept for 30 min at room temperature. BSA was replaced with phalloidin TRITC (Sigma-Aldrich Co.) diluted in a ratio of 1:600 and DAPI in a dilution of 1:1000 (Sigma-Aldrich Co., Saint Louis, MI, USA) and kept at room temperature for 40 min. After staining, scaffolds were raised four times with PBS and gently shaken.

All scaffolds were kept in PBS overnight before analysis. Vectashield mounting was used to prepare samples for microscopy. A 40× lens was used for all experiments (EC Plan-Neofluar 40×/1.30 Oil DIC M27, Carl Zeiss, Oberkochen, Germany) with excitation wavelengths set to 488 and 405 nm and pinhole set to 45 μm.

## 3. Results

### 3.1. Analysis of Perovskite Particle Size

Sizes of the perovskite particles synthesized in this work were measured by a ZetaSizer Nano S. Each sample was measured trice and experiments were doubled in order to obtain statistically relevant data. Results revealed sizes of non-substituted perovskites of 424.4 ± 13.0 nm and 484.8 ± 14.1 nm for substituted material. Analyzing sizes of particles also showed the presence of a residual phase with sizes around 5000 nm in amounts ranging from 1% to 5% of the whole volume (Figure 1).

### 3.2. X-ray Diffraction of Perovskite Particles

X-ray diffraction was performed on non-substituted and substituted perovskite material with scanning range between of 5° and 65°. The obtained spectrum was analyzed by Match 2 software (Match! Crystal Impact, Bonn, Germany) and fitting by the Rietveld method (Full Prof) between experimental and theoretical data was used with the Crystallography Open Database (COD-07-01-2015) as reference data (Figure 2). Results showed reflection conditions for Na_2_La_2_TiO_3_O_10_ and Ag_0.3_Na_1.7_La_2_Ti_3_O_10_ to be *h + k + l = 2n*, which indicates tetragonal crystalline structure of both obtained compounds [44]. Because of differences between atom sizes of Na^+^ (0.98 Å) and Ag^+^ (1.13 Å), substitution to Na_2_La_2_TiO_3_O_10_ did not result in changes in crystalline structure due to tetragonal structure limitations. Therefore, unit cell parameters for both compounds were calculated as Na_2_La_2_TiO_3_O_10_ (a = 3.8331 Å, c = 28.6428 Å and z = 2); Ag_0.3_Na_1.7_La_2_Ti_3_O_10_ (a = 3.8382 Å, c = 28.6545 Å and z = 2), which is in agreement with previously reported data [29]. Because of atom size differences and small amounts of silver added to the system, very small differences between the two obtained spectra were observed.

### 3.3. Electrospinning Process Optimization ad SEM Analysis

All scaffolds were electrospun under these same conditions (25 °C) and humidity (of approximately 60%) with a polymer flow of 1.2 mL/h and electric potential difference of 13.7–14.0 kV. Distance between needle and collector was set to 12–13 cm. Initial quality of fibers was evaluated on the optical microscope. Fiber diameters were measured in 100 points on every picture of magnification of 3000×. All measurements were done in ImageJ software. Average of fiber diameters were in the nanoscale, under 1000 nm, therefore classifying them as nanofibers and beneficial for tissue engineering [30]. The average fiber diameters decreased from 610 ± 15 nm (PLGA-1P) to 408 ± 80 nm (PLGA-15P) with the addition of perovskite material to the polymeric matrix. The increase of diameter between the pure polymeric matrix (PLGA-0P) and incorporated material (PLGA-1P) was not statistically significant (Figure 3).

However, the decrease in diameter between PLGA-10P and PLGA-15P was insignificant; nevertheless, the change of diameter between the incorporated and non-incorporated fibers remained detectable, which was likely due to the increased conductivity of the polymer solution [45]. Distribution of fiber diameters also showed a shift of median value towards smaller diameters with an increase of perovskite concentration (Figure 4). All histograms showed a Gauss distribution of fiber diameters.

### 3.4. Mechanical Properties and Contact Angle of Electrospun Nanofibers

The scaffolds prepared by electrospinning were evaluated with a universal tensile machine. Examples of stress–strain curves and values of Young’s modulus, ultimate tensile strain and ultimate tensile stress are presented in Figure 5. Calculation of Young’s modulus showed similar values between PLGA-0P and PLGA-1P of 130.8 ± 7.5 and 139.1 ± 7.3 MPa, respectively. An increase of perovskite content caused a decrease in Young’s modulus from 94.5 ± 3.5 MPa for PLGA-5P, to 83.7 ± 6.6 MPa for PLGA-10P and 74.6 ± 7.2 MPa for PLGA-15P.

Ultimate tensile strain and ultimate tensile stress of the constructed scaffolds was also found to decrease with increasing amount of perovskite. Maximal strain was observed for PLGA-0P scaffolds (254.4 ± 53.0%). Immobilization of polymer chains by particles decreased the Young’s modulus and maximum strain of scaffolds to 94.8 ± 4.0% for PLGA-1P and 56.6 ± 9.7% for PLGA-15P. Maximum stress that could be applied to scaffolds before breakage was shown to be 4.5 ± 0.2 MPa for PLGA-0P and decreased to 2.3 ± 0.5 MPa for PLGA-15P. Hydrophobicity of the obtained scaffolds did not show significant change and ranged from 134.3 ± 0.7 for PLGA-0P to 139.6 ± 1.2 for PLGA-15P (Table 1).

### 3.5. FTIR Analysis of Polymeric Perovskite-Incorporated Scaffolds

Fourier transform infrared spectroscopy was performed on polymeric scaffolds cut to circular samples of diameters of approximately 5 mm. Scanning in the range of 400–4000 cm^−1^ was done using the transmittance method, and results of measurement were later analyzed by Origin Pro 8.1 software (Origin Lab, Northampton, MA, USA). Results revealed no significant changes between PLGA-0P and other scaffolds. Bands observed near 1759 cm^−1^ were present due to stretching of carbonyl groups from the polymeric matrix. Similarly, bands around 2944 and 2993 cm^−1^ were present due to stretching vibrations of −CH, −CH_2_ and −CFH_3_ groups. Peaks in the range of 1200–1500 cm^−1^ (1272, 1383 and 1453 cm^−1^) represented the deformation vibrations (−CH, −CH_2_ and −CH_3_ groups present in the polymer). Peaks in the range of 1100–1300 cm^−1^ were connected to stretching vibrations of ester bond (C–O) and wagging vibrations of −CH_2_ and −CH at 1371 and 1150 cm^−1^, respectively (Figure 6). Perovskite-incorporated polymeric fibers in the nanoscale were proposed to be an inorganic phase. The peaks characteristic for perovskite phase were not visible in the FTIR spectrum.

### 3.6. Anti-Microbial Characteristic of Synthesized Perovskite and Electrospun Fibers

Anti-microbial activity was measured for non-substituted and substituted perovskite materials as well as for fibrous scaffolds with various concentrations of perovskite particles. Four Gram-negative (*K. pneumonia* (DM4299), *E. coli* (16027R), *P. aeruginosa* (23155 and 4299)) and four Gram-positive (*S. saprophyticus* (15305, BAA750, 49907, 49453)) strains as well as two species of fungi (*C. albicans* (CA1976, CA2672)) were used in this test to evaluate anti-microbial properties. Bacteria-free area was measured as the zone of inhibition for all microorganisms. Substituted perovskite powder Na_2_La_2_TiO_3_O_10_ (Perovskite S) was used as positive control, while PLGA-0P was used as negative control. Non-substituted powder Ag_0.3_Na_1.7_La_2_Ti_3_O_10_ (Perovskite NS) was also tested for anti-microbial presence in order to estimate if newly synthetized materials released silver ions. A summary of zone of inhibition of all materials is presented in Table 2. Na_2_La_2_TiO_3_O_10_ perovskite powder showed no anti-microbial properties, similarly to PLGA-0P scaffolds. Substituted material Ag_0.3_Na_1.7_La_2_Ti_3_O_10_ showed good anti-microbial properties against Gram-negative bacterial strains with a range of inhibition of 0.7–1.9 cm, whereas Gram-positive bacterial strains showed a zone of inhibition of 0.4–0.9 cm. Perovskite also showed positive results against two fungi species with a zone of inhibition of 0.5–0.7 cm. PLGA-1P and PLGA-5P showed no zone of inhibition, as shown in Figure 7, for all strains where when PLGA-10P was effective against *P. aeruginosa* (PA23376) and *S. saprophyticus* (SSBAA750) strains. PLGA-15P effectively inhibited the bacterial growth for *K. pneumoniae* (DM4299), *P. aeruginosa* (PA23376) and three out of four strains of *S. saprophyticus* (SS49907, SSBAA750 and SS49453).

### 3.7. Proliferation and Morphology of HDF

Proliferation AlamarBlue^®^ assay was performed 2, 4, 6 and 8 days after cells was seeded on scaffold surfaces, as shown in Figure 8. The assay was performed every two days, and scaffolds were kept at incubation environment at 37 °C in between experiments. A calibration curve was used as described above to ensure the correct number of calculated cells. Each well was seeded with 10,000 cells, and the number of cells was calculated at every stage of the proliferation study. The number of cells proliferated on the surface of PLGA-0P after 8 days showed values similar to the positive control TCP group (115.348 ± 7.731 and 128.152 ± 6.705, respectively). Perovskite-incorporated fibers showed an increase of cell number by 62–68% for all examined scaffolds between day 2 and day 4. Cells proliferated steadily between day 6 and 8 and showed an increase of cell number by 25, 14, 10, 30 and 19% for PLGA-0P, PLGA-1P, PLGA-5P, PLGA-10P and PLGA-15P, respectively. When comparing initial number of cells with amount, after 8 days, of cells seeded on TCP, they grew by 980%, 1103%, 947%, 842%, 859% and 1007% for PLGA-0P, PLGA-1P, PLGA-5P, PLGA-10P and PLGA-15P, respectively.

### 3.8. Immunofluorescence of Cellular Cytoskeletal Marker

In order to analyze the morphology of cells and their interaction with perovskite rich scaffolds, DAPI (4’,6-diamidino-2-phenylindole) and phalloidin staining was conducted to identify F-actin cytoskeleton in cells. Actin is a major cytoskeletal marker used in tissue engineering to identify normal cellular morphology. DAPI binds with nucleus regions rich in A–T bonds within the DNA helix and is able to pass intact through the cell membrane. With excitation wavelengths of 358 nm and emission maximum of 461 nm, it is usually detected by a blue filter [46]. Emission radiation was measured in this test by a detector set up to 490 nm. Phalloidin was used to stain F-actin protein within the HDF cell cytoskeletons. Prepared cells were excited by laser wavelength of 488 nm, and emitted light was detected by 555 nm detector [47]. Figure 9 represents pictures obtained by confocal microscopy of cells present on scaffolds. Staining and imaging were performed on day 9 of the culture test using a laser scanning confocal microscope (LSCM). Results revealed very good interaction of cells on the TCP sample with presence of fibrous F-actin chains, which are typical for human dermal fibroblasts cells [48]. All scaffolds incorporated with perovskite showed similar results with cells tightly attached to the fibrous construct and presence of fibroin F-actin structure, which ensured good communication between cells. We noticed that F-actin produced on cells on TCP were fibrous and randomly oriented when protein presence on scaffolds was arranged toward one direction, which was connected with the arrangement of fibrous meshes. PLGA-0P showed slightly lower presence of cells and F-actin when compared with PLGA-15P and TCP surfaces. Furthermore, PLGA-5P showed complete coverage with cells similarly to how it happened with TCP samples, but with oriented fibers with F-actin along the fibers. These results are consistent with our previous findings [49].

## 4. Discussion

There is a great need for new anti-microbial materials to be developed and safely implemented in wound management strategies. New materials are especially needed due to the increasing number of antibiotic resistant bacterial strains and number of burn and acute wound cases. Novel anti-microbial materials in combination with nanofibrous structures providing mechanical properties and unique microstructure might be an interesting alternative to traditional wound dressings. The aim of this study was to fabricate nanofibrous biodegradable scaffolds incorporated with synthesized perovskite particles. XRD analysis of synthesized material showed the existence of a tetragonal structure of Na_2_La_2_TiO_3_O_10_ and Ag_0.3_Na_1.7_La_2_Ti_3_O_10_. Those results were in accordance with reported previously work [28,29]. The size of the particles obtained from solid state synthesis was relatively small, such that the incorporation of those particles into a polymeric matrix was also presented in the analysis of mechanical and morphological changes. For tissue engineering applications, electrospun scaffolds with fiber diameters less than 1000 nm are often referred to as nanostructured or nanofiber scaffolds [30]. Previous studies by our group have shown such fine fibrous architectures to be beneficial for the growth of cells from several organ tissues, including skin [45,50,51,52,53,54,55]. In this study, the morphology of electrospun fibers illustrated by scanning electron microscopy showed optimal fiber diameters in a similar range, which is known to support cell growth with a tendency to finer fiber diameters with increasing amounts of anti-microbial perovskite, which is more preferable for HDF cells [56]. Changes in fiber diameter when incorporated with small amount of the additives is a known phenomenon that may occur because of decreases of viscosity and increases of conductivity of the polymeric solution [45,57].

In particular, the presence of silver-based additives are known to increase the electric charge and conductivity of the polymer solution and reduce the fiber drastically [58,59]. This has been attributed to the higher stress on the whipping jet of the metal-rich portion during the phase separation, which created subsidiary jets during the electrospinning process resulting in the formation of thinner nano-nets along with the main fibers. Therefore, it is likely that increasing silver-based perovskite concentration in our study reduced the fiber diameter [59]. As a result, they have the potential to largely influence cellular behavior and in vivo performance [49,51]. In our study, it is likely that the thinning of nanofibers with increasing concentration of perovskite was due to increased conductivity of the solution. This is an interesting phenomenon and can be used to tailor-make scaffolds for tissue engineering.

Young’s modulus of obtained fibers decreased from 130.8 to 74.6 MPa for PLGA-0P and PLGA-15P, respectively. Our results suggest that presence of perovskite in PLGA mobilized the fiber. Values of ultimate tensile stress and ultimate tensile strain decreased from 4.5 MPa for PLGA-0P to 2.3 MPa for PLGA-15P and from 264.4% to 56.6% for PLGA-0P to PLGA-15P, respectively. Mechanical properties of incorporated polymeric nanofibers were compatible with the mechanical properties of human skin with an ultimate stress maximum of 3.0 ± 1.5 MPa [60,61]. Surface contact angle remained unchanged with relatively small variations between samples, which suggested that perovskite crystals may be isolated by the presence of polymeric shielding. FTIR analysis did not reveal changes in prepared materials due to the lack of chemical interaction between ceramic particles and the polymeric matrix and proved a shielding effect between those two phases. Tests of anti-microbial properties of prepared powders and perovskite incorporated fibers revealed anti-microbial activity of synthesized material and some of the polymeric constructs presented in this work. Non-substituted perovskite materials showed no anti-microbial properties against all four Gram-positive and Gram-negative bacterial strains and two species of fungi. PLGA-0P, as well as perovskite-incorporated scaffolds PLGA-1P and PLGA-5P, showed no anti-microbial activity. Substituted perovskite material showed good anti-microbial properties comparable with the efficiency of silver nanoparticles against those same bacterial strains [62]. Scaffolds with increased amounts of perovskite PLGA-10P showed some anti-microbial activity against *P. aeruginosa* (PA23376) and *S. saprophyticus* (SSBAA750), while PLGA-15P, with the highest concentration of perovskite, showed effectiveness against *K. pneumonia* (DM4299), *P. aeruginosa* (PA23376) and *S. saprophyticus* (SS49907, SSBAA750, SS49453). Those results indicate that despite good anti-microbial properties of perovskite powder alone, the concentration of perovskite material within polymeric fibers is too low to create good resistance for bacteria and fungi. Furthermore, the size of perovskite particles, possible agglomeration, interaction between polymer and perovskite and release rate of silver ions from the crystal structure are factors that also limit anti-microbial properties. The effectiveness of PLGA-10P and PLGA-15P against these same bacteria strains and increases of the size of the bacteria free area with the increase of perovskite content also suggests that concentration is likely the key factor in anti-microbial properties.

Electrospun materials showed very good interaction with human dermal fibroblasts with values comparable to a pure polymeric matrix, which suggests that perovskite used in this work does not show any cytotoxicity to human skin cells. TCP surfaces are a good positive control to compare the growth of cells on scaffolds, as these surfaces are primarily designed for tissue culture. As a result, cell growth on electrospun scaffolds mostly remains lower than this positive control of TCP, as shown in several of our previous studies [49,50,51,63,64]. Yet, our study shows that cell growth on our scaffolds, although slower than positive control TCP, do support the growth and expression of essential proteins such as F-actin in dermal fibroblasts. Furthermore, unlike silver nanoparticles, increases of local concentrations of perovskite did not show increases of cytotoxicity [65]. Our results show possible applications of perovskite-incorporated nanofibers as wound dressings with concentrations of perovskite more than 15% (w/w) without a decrease in cytotoxicity. Such scaffolds may find significant potential in the area of skin and gynecological tissue engineering where the anti-microbial meshes have been indicated as an emerging need for designing the next generation of surgical implants.

## 5. Conclusions

The aim of this study was to produce and evaluate possible applications in medicine of perovskite-incorporated, biodegradable nanofibrous materials. Synthesis and analysis of two layered perovskites and their anti-microbial properties were evaluated in order to ensure that anti-microbial properties of this novel material are induced by the presence of silver ions in the lattice. It is well known that perovskites incorporate extremely well within nanofibers fabricated by electrospinning and has been reported by several groups [37,38,39,40,41]. This makes electrospinning an attractive strategy for fabricating such anti-microbial nanofibrous scaffolds using Ag-based perovskites. However, our study is the first to explore the application of such scaffold for wound healing tissue engineering applications with anti-microbial properties. Mechanical properties of the produced materials showed similarity to native skin tissue and smaller sizes of the fibers with increasing perovskite content increased positive responses from human dermal fibroblasts. The antibacterial activity was measured against *E. coli*, *P. aeruginosa*, *K. pneumoniae*, *S. saprophyticus* and *C. albicans*, and scaffolds PLGA-10P and PLGa-15P were found to inhibit bacteria growth, dependent on perovskite concentration. Anti-microbial properties of pure perovskite powder against both bacteria and fungi were found to be comparable with other known anti-microbial agents; therefore, these materials were found to be effective likely due to the release of silver ions. Polymeric meshes with perovskite as anti-microbial agents were found to be an effective candidate as novel materials for wound dressings. Such nanomaterials may find potential in diverse tissue engineering biomedical applications such as urogynecology. While our work is the first report of perovskite-based nanofibers as tissue engineering scaffolds, further studies are needed to study the distribution of perovskites within the nanofibers and the kinetics of ion release from these scaffolds that enable such anti-microbial properties. Our future studies would be aimed at understanding the cell–material interactions and the foreign body response associated with perovskite-based nanofibers using appropriate in vivo models.

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
