# Peer review of "Design of Novel Perovskite-Based Polymeric Poly(l-Lactide-Co-Glycolide) Nanofibers with Anti-Microbial Properties for Tissue Engineering"

_nanomaterials, 2020, doi:10.3390/nano10061127_

Round 1
Reviewer 1 Report
This paper needs major revision with more experiments, as commented below:
- Authors need to explain the weakness of the nanofiber scaffolds in the introduction part, referencing below refs that tried to address this issue:
- Chen H et al. Direct Writing Electrospinning of Scaffolds with Multidimensional Fiber Architecture for Hierarchical Tissue Engineering. ACS Appl Mater Interfaces. (2017) / Asencio, Ilida Ortega et al. A methodology for the production of microfabricated electrospun membranes for the creation of new skin regeneration models. J Tissue Eng, 2018; 9: 2041731418799851.
- They should measure the release of ions from the nanofibers
- Scheme 1 & 2 should be combined
- They have to show the presence of NPs within fibers by TEM
- Authors need to quantify the zone of inhibition study.
- Authors need to reason why all the samples have lower cell viability than TCP.
- Some of the key recent works on antibacterial drug releases using different types of biomaterials (as shown below) should be cited to feedback recent research trend in this field.
- Gimeno et al, A controlled antibiotic release system to prevent orthopedic-implant associated infections: An in vitro study, Eur J Pharm Biopharm. 2015; 96: 264–271 / Moon et al. Near-infrared laser-mediated drug release and antibacterial activity of gold nanorod–sputtered titania nanotubes. Journal of Tissue Engineering, 2018; 9: 2041731418790315 / Mountziaris et al, A rapid, flexible method for incorporating controlled antibiotic release into porous polymethylmethacrylate space maintainers for craniofacial reconstruction, Biomater Sci, 2016, 1.
Reviewer 2 Report
The manuscript entitled "Design of novel perovskite based polymeric poly(L-lactide-co-glycolide) nanofibers with antimicrobial properties for tissue engineering" (Ref: nanomaterials-791791) reports a novel nanofibers of poly(L-lactide-co-glycolide) blended with perovskite nanoparticles for its use in several biomedical application areas.
My overall impression is positive and the manuscript does deserve publication after deep revision.
In general, authors must do a careful revision related to bibliography. There are some mistakes and format of references must be standarized. In addition, they have to review the writting of bacteria. The most important corrections that authors should carry out are:
- Unify Results and Discussion sections.
- Electrospinning Process (3.3): explain why fiber diameters decreased with addition of perovskite material to polymeric matrix.
- Mechanical Properties (3.4):
- Ultimate Tensile Stress values for samples, PLGA-5P, PLGA-10P, and PLGA-15P, shown in graph do not correspond with those detailed in table.
- Provide more detailed explanations in discussion supported by references.
- Remove graphs B, C, and D in Figure 5, those do not provide relevant information,
Round 2
Reviewer 2 Report
Dear authors, I really appreciate you have fully addressed my concerns, and the manuscript is now satisfactory for publication in this journal.
Author Response
Reviewer 2 Comments: Dear authors, I really appreciate you have fully addressed my concerns, and the manuscript is now satisfactory for publication in this journal.
Author Response: We thank the reviewer for their taking the time to review our revisions. We are delighted to note the top scores they have given us in all categories and have approved our manuscript for publication.